# The Use of a New Dedicated Electrocautery Lumen-Apposing Metal Stent for Gallbladder Drainage in Patients with Acute Cholecystitis

**DOI:** 10.3390/diagnostics13213341

**Published:** 2023-10-30

**Authors:** Luca Brandaleone, Gianluca Franchellucci, Antonio Facciorusso, Jayanta Samanta, Jong Ho Moon, Jorge Vargas-Madrigal, Carlos Robles Medranda, Carmelo Barbera, Francesco Di Matteo, Milutin Bulajic, Francesco Auriemma, Danilo Paduano, Federica Calabrese, Carmine Gentile, Marco Massidda, Marco Bianchi, Luca De Luca, Davide Polverini, Benedetta Masoni, Valeria Poletti, Giacomo Marcozzi, Cesare Hassan, Alessandro Repici, Benedetto Mangiavillano

**Affiliations:** 1Digestive Endoscopy, IRCCS Humanitas Research Hospital, 20089 Rozzano, Milan, Italy; davide.polverini@humanitas.it (D.P.); benedetta.masoni@humanitas.it (B.M.); valeria.poletti@humanitas.it (V.P.); giacomo.marcozzi@humanitas.it (G.M.); cesareh@hotmail.com (C.H.); alessandro.repici@hunimed.eu (A.R.); 2Department of Biomedical Sciences, Humanitas University, 20072 Pieve Emanuele, Milan, Italy; 3Gastroenterology Unit, Department of Biomedical Science, Foggia University Hospital, 71122 Foggia, Puglia, Italy; antonio.facciorusso@virgilio.it; 4Gastroenterology PGIMER, Post Graduate Institute of Medical Education and Research, Chandigarh 160012, India; dj_samanta@yahoo.co.in; 5Digestive Disease Center and Research Institute, Department of Internal Medicine, SoonChunHyang University School of Medicine, Bucheon 31538, Republic of Korea; jhmoonsch@gmail.com; 6Department of Gastroenterology and Endoscopy, Hospital Enrique Baltodano Briceño, Liberia 50101, Costa Rica; gastrocr@gmail.com; 7Instituto Ecuatoriano de Enfermedades Digestivas—IECED, Gastroenterology Av Abel Romero Castillo, Guayaquil 090505, Guayas, Ecuador; carlosoakm@yahoo.es; 8Digestive Endoscopy Unit, Fondazione Policlinico Universitario Campus Bio-Medico, Via Alvaro del Portillo 200, 00128 Rome, Rome, Italy; carmelo.barbera@gmail.com; 9GIUnit, Digestive Diseases, Campus Bio Medico University, 00128 Rome, Rome, Italy; f.dimatteo@policlinicocampus.it; 10Digestive Endoscopy, Fatebenefratelli Isola Tiberina—Gemelli Isola, 00186 Rome, Rome, Italy; bulajic.milutin@gmail.com; 11Gastrointestinal Endoscopy Unit—Humanitas Mater Domini, 21100 Castellanza, Varese, Italy; francesco.auriemma.1987@gmail.com (F.A.); danilo.paduano@materdomini.it (D.P.); federica.calabrese@materdomini.it (F.C.); carmine.gentile@materdomini.it (C.G.); 12Gastroenterology and Digestive Endoscopy Unit, Mater Olbia Hospital, 07026 Olbia, Sassari, Italy; marco.massidda@materolbia.com; 13Policlinico Casilino Hospital, 00169 Rome, Rome, Italy; dr.marcobianchi@gmail.com; 14Endoscopic Unit, ASST Santi Paolo e Carlo, 20142 Milan, Milan, Italy; lucadeluca1210@gmail.com

**Keywords:** new dedicated lumen-apposing metal stent (LAMS), acute cholecystitis, therapeutic endoscopic ultrasonography

## Abstract

**Aims:** Lumen-apposing metal stents (LAMSs) in ultrasonography-guided gallbladder drainage (EUS-GBD) have become increasingly important for high-risk surgical patients. Our study aims to evaluate the technical and clinical success, safety, and feasibility of endoscopic ultrasonography-guided gallbladder drainage using a new dedicated LAMS. **Methods:** This is a retrospective multicenter study that included all consecutive patients not suitable for surgery who were referred to a tertiary center for EUS-GBD using a new dedicated electrocautery LAMS for acute cholecystitis at eight different centers. **Results**: Our study included 54 patients with a mean age of 76.48 years (standard deviation: 12.6 years). Out of the 54 endoscopic gallbladder drainages performed, 24 (44.4%) were cholecysto-gastrostomy, and 30 (55.4%) were cholecysto-duodenostomy. The technical success of LAMS placement was 100%, and clinical success was achieved in 23 out of 30 patients (76.67%). Adverse events were observed in two patients (5.6%). Patients were discharged after a median of 5 days post-stenting. **Conclusions**: EUS-GBD represents a valuable option for high-surgical-risk patients with acute cholecystitis. This new dedicated LAMS has demonstrated a high rate of technical and clinical success, along with a high level of safety.

## 1. Introduction

Acute cholecystitis represents one of the most frequent causes of hospital admission in Western countries. In the US, acute cholecystitis has an incidence of about 200,000 people per year [1], with more than 700,000 hospitalizations being attributed to acute cholecystitis as the main diagnosis in 2018 [2]. The MICOL study, an Italian population-based cross-sectional study, evidenced an incidence rate of 0.67% per year with respect to the development of gallstone disease [3]. Cystic duct obstruction by calculi is the pivotal trigger for 90–95% of acute cholecystitis cases. Approximately 5–10% experience gallbladder inflammation without calculi, known as acalculous cholecystitis. Obstruction by calculi increases intraluminal pressure, inflaming the gallbladder wall, followed by bacterial infections, typically with enteric organisms like *Escherichia coli*, *Klebsiella* spp., and *Staphylococcus* spp. Clinically, acute cholecystitis exhibits right upper quadrant abdominal tenderness, fever, nausea, and a distinctive Murphy’s sign, where pain halts inspiration on right upper quadrant palpation [1]. Jain et al. revealed Murphy’s sign’s sensitivity and specificity as 62% and 96% [4]. Elevated CPR levels and white blood cell (WBC) counts are typically observed in laboratory exams. However, clinical presentation and blood exam results lack definitive positive or negative likelihood ratios for diagnosing acute cholecystitis [1]. Ultrasonography (US) is the primary modality for gallbladder and biliary tree imaging due to its accessibility, cost-effectiveness, and non-X-ray nature. Ultrasonographic signs include cholecystic lithiasis, US Murphy’s sign, pericholecystic fluid, and gallbladder wall thickening >3 mm. US sensitivity and specificity for acute cholecystitis are 81% and 80%, respectively [5]. Computed tomography (CT) is the second most commonly used technique, and it can be used to indicate distended gallbladder walls, wall thickening, and pericholecystic fluid. Gallstone visibility depends on composition and CT slice thickness. Roughly 20% of gallstones remain invisible due to their similar attenuation to bile [6]. Kiewiet et al.’s systematic review and meta-analysis of imaging’s role in acute cholecystitis revealed a CT sensitivity of 94% and specificity of 59% [5]. In cases of gangrenous cholecystitis, the primary imaging findings encompass irregular thickening of the gallbladder wall, diminished contrast enhancement of the gallbladder wall (referred to as the interrupted rim sign), the presence of gas within the gallbladder lumen, the existence of intraluminal flaps and membranes, or the development of a pericholecystic abscess. The interrupted rim sign demonstrates a sensitivity of 76% and an impressive negative predictive value (NPV) of 95% [7]. Magnetic resonance imaging (MRI) and MR cholangiopancreatography offer viable alternatives, as they are capable of detecting gallstones, gallbladder wall thickening (greater than 3 mm), gallbladder distension (exceeding 40 mm), wall edema, and the presence of pericholecystic or perihepatic fluid [8]; sensibility and specificity is about 88% and 89%, respectively [9]. Furthermore, MRI serves as a valuable tool for identifying complications such as gangrenous, emphysematous, or perforated cholecystitis while simultaneously ruling out the presence of choledocholithiasis in cases of acute cholecystitis [10,11]. A nuclear medicine diagnostic test is employed to achieve a comprehensive visualization of the gallbladder and biliary tree. This test, known as hepatobiliary scintigraphy or hepatic iminodiacetic acid scan (HIDA scanning), utilizes a technetium-labeled analogue of iminodiacetic acid (radiotracer). It capitalizes on the intravenous administration of a sub-analgesic dose of morphine, which induces the contraction of Oddi’s sphincter, diverting incoming bile to the gallbladder. Patients are required to observe a fasting period lasting between 4 to 6 h. Approximately 30 min after the administration of the radiotracer, assuming the cystic duct is patent, gallbladder filling should become visually discernible [12,13]. The inability to visualize the gallbladder confirms cystic duct obstruction. Kiewet et al. [5] demonstrated a sensitivity and specificity of 96% and 90%, respectively, for hepatobiliary scintigraphy in diagnosing acute calculous cholecystitis. In summary, the diagnosis of acute cholecystitis is predicated based on a synthesis of clinical, laboratory, and radiological findings. The diagnostic criteria for acute cholecystitis are delineated by the Tokyo Guidelines [14], which are detailed in Figure 1.

Cholecystectomy is the treatment of choice for calculous acute cholecystitis [15]. Early laparoscopic cholecystectomy has been adopted as the treatment of choice in cases of acute cholecystitis. An approach is considered an early surgical approach when a cholecystectomy is performed within 24, 48, 72 h, or 96 h from either hospital admission within one week from the onset of the symptoms [16]. Post-operative complications are mainly represented by abdominal wall or intra-abdominal bleeding and wound infections. Many studies and reviews have evaluated outcomes and rates of post-operative complications related to early vs. delayed cholecystectomy. Early laparoscopic cholecystectomy is characterized by a lower rate of post-operative complications, defined as events occurring within 75 days. Furthermore, the mean hospital length values and costs associated with early laparoscopic cholecystectomy are significantly lower than that of delayed laparoscopic surgery. However, the use of an early non-operative approach with a delayed laparoscopic cholecystectomy was assessed in elderly people (age > 65 years old). Hazzan et al. evidenced how early surgical approaches are safe and acceptable for morbidity and mortality in this population, and the seems to be true for younger people [17,18].

Among patients ineligible for surgery, endoscopic ultrasonography-guided gallbladder drainage (EUS-GBD) has been gaining more and more importance in the last 10 years [19]. More and more patients will become not suitable for surgery due to the increasing prevalence of the elderly population. It has been predicted that people older than 65 years old will represent almost the half of the entire population in Europe by 2050. In this context, approaches such as EUS-GBD or endoscopic transpapillary gallbladder drainage (ET-GBD) or the percutaneous approach (PT-GBD) may represent valuable conservative therapeutic options. Recent studies have evidenced the superiority (in terms of clinical success) of EUS-GBD and TP-GBD, with the former having a clinical success rate near to 90% and the latter having a clinical success rate of about 80% [20]. In this context, choosing between EUS-GBD and percutaneous transhepatic gallbladder drainage (PT-GBD) is a heavily debated topic. A recent meta-analysis showed that technical and clinical success were achieved, respectively, in 98% and 95% of patients, with the rate of adverse events being 14.8%. Another meta-analysis evidenced the superiority of EUS-BD drainage over the PT-GBD approach only when cautery-enhanced LAMSs are used [19]. In the DRAC1 study (a randomized controlled trial), the EUS-GBD showed a significant low rate of readmission after 30 days and a low adverse event at 1 year of follow-up. All of the studies performed to evaluate the performances of EUS-GBD were conducted using Axios (Boston Scientific Medical Corporation, Marlborough, USA), the only commercially available LAMS [21]. In terms of follow-up in the context of EUS-GBD drainage, two distinct endoscopic approaches are currently employed. One option involves performing peroral cholecystoscopy at 4 or 6 weeks post-gallbladder drainage to ensure complete gallstone clearance. Simultaneously, plastic stents are placed to maintain long-term patency of the cholecysto-gastric fistula. An alternative endoscopic approach involves leaving a lumen-apposing metal stent (LAMS) in situ, demonstrating a remarkable long-term patency rate of 86% and a delayed adverse event occurrence of 7% [22,23]. Spaxus (Taewoong Medical Co., Gimpo, Republic of Korea) are newly dedicated LAMSs produced by Taewoong, and only a few examples of their use in GBD are available in the literature [24,25]. The aim of this study is to assess both the clinical and technical success and the rate of adverse events of EUS-GBD in high-volume centers via the Hot-Spaxus LAMS.

## 2. Patients and Methods

This is retrospective multicenter study included all consecutive high-surgical-risk patients who underwent EUS-GB with the use of an electrocautery LAMS (Hot-Spaxus, Taewoong Medical Co., Gimpo, Republic of Korea) for acute cholecystitis at eight different centers. Patient recruitment occurred from 14 July 2020 to 1 February 2023. The study protocol was approved by the Ethics Committee of each participating institution. A multidisciplinary Team evaluated each clinical scenario, and once cholecystectomy (LAC) was excluded, EUS-GBD was proposed and accepted as a valuable and alternative conservative approach instead of surgery to decompress the biliary tree and reduce the rate of life-threating AC-related conditions such as perforation and peritonitis. The enrollment algorithm is shown at Figure 2.

### 2.1. Outcomes Definition and Measurements

The primary and secondary outcomes of the present study was to analyze the technical and clinical success of EUS-GBD, respectively. Technical success was defined as the correct placement of the LAMS in either the duodenum or the stomach. The incidence of Adverse events (AEs) was recorded, and the severity of AEs was classified according to the American Society of Gastrointestinal Endoscopy Lexicon [26]. Post-procedure adverse events were considered to be events occurring within 14 days, and late AEs were considered to be those occurring 14 days after the endoscopic procedure. Clinical success was defined as the resolution of the inflammatory process, as evidenced by CRP (C-reactive protein) levels <3 mg/dL and a white blood cell count (WBC) below <10,000/mm^3^ at five days after the endoscopic procedure, along with the absence of clinical signs associated with acute cholecystitis.

#### 2.1.1. Inclusion Criteria

All the patients included in the study were able to understand the protocol and sign an informed consent. All patients had a diagnosis of some grade of acute cholecystitis (diagnosed according to the latest update of the Tokyo guidelines) [14]. Patients had to be older than 18 years and must have been declared not eligible for surgery because of their age, comorbidities, or overall health status.

#### 2.1.2. Exclusion Criteria 

Eligibility for surgery; impossibility to understand or sign the informed consent form; technical or clinical impossibility to perform EUS-GBD.; presence of percutaneous gallbladder drainage, ascites, pregnancy, coagulopathy, impossibility to withdraw antiplatelets or conduct anticoagulant therapy according to the latest guidelines [27,28], and allergies prohibiting EUS GB drainage.

### 2.2. Study Device

Hot-Spaxus was utilized for all procedures (Figure 3). This device consists of braided nitinol fully covered with silicone and at the ends flexible flanges with different lengths. The flares provide accommodative apposition regardless of the wall thickness and have a channel in which a 0.035-inch guidewire can be preloaded. The 3 available stents with diameters and bodies of 8 × 20 mm, 10 × 20 mm, and 16 × 20 mm, respectively, were all delivered through a 10F delivery catheter and had flange diameters of 23, 25, and 31 mm, respectively. The electrocautery tip allows for the passage of the catheter into the target structure without prior tract dilation through a cutting current of 80–120 watts set on pure cut mode (Figure 4 and Figure 5). The choice of the LAMS was evaluated according to the distance from the gallbladder to the duodenum or stomach in the endoscopic ultrasound, and the choice between the transgastric and transduodenal approach depended on the most comfortable and stable endoscopic position.

### 2.3. Procedure

The EUS-GBD drainage procedures were performed by expert endoscopists under either deep sedation or general anesthesia, depending on the anesthesia protocol adopted at each tertiary center. Before the procedure, all patients received antibiotic prophylaxis tailored to the local antimicrobial resistance patterns observed at the participating centers. In all documented procedures, a therapeutic linear echoendoscope, carefully selected from the available types that had received regulatory approval, was utilized. Once the target structure was properly identified from the stomach or duodenum, Doppler examination was used to rule out interposed vessels and identify the best site for stent insertion. The EC-LAMS was placed at the discretion of each endosonographer using a freehand technique or after the puncture of the target organ or cavity with a standard 19-gauge FNA needle, followed by guidewire placement. Proximal flange release was performed under endoscopic or EUS views by using the “intra-channel release” technique, in which the flange is released inside the operative channel of the echoendoscope and then pushed outside by pulling back the echoendoscope. The LAMSs employed in these procedures were Hot-Spaxus stents, available in calibers of 8 mm, 10 mm, and 16 mm, with a standard length of 20 mm.

### 2.4. Data Analysis

All the data pertaining to the performed procedures were collected and sorted into a dedicated dataset with 58 different variables for each patient demographic parameters. Categorical data are described using frequency and composition ratios. Continuous data are described using mean, standard deviation, maximum, minimum, and median. The continuous quantitative variables were analyzed via univariate analyses, and multivariate analyses were performed only when a significant relationship was found in the univariate ones. The data analyses were performed using R Statistical Software (version number 23.0, IBM, Chicago, IL, USA) (Foundation for Statistical Computing, Vienna, Austria), and significance was established at the 0.05 level (2-sided).

## 3. Results

### Baseline Characteristics

Between June 2014 and February 2023, fifty-four consecutive patients were underwent EUS-GBD for acute cholecystitis. Of these, the mean age was of 76.48 years (standard deviation 12.6 years). Of the patients, 42.6% were male. CBD stones were present in 14 (25.9%) patients; hence, endoscopic retrograde cholangiopancreatography (ERCP) was performed in the same sessions for five patients (9.3%). Among the patients who underwent EUS-GBD, seven (13.0%) had a personal history with gallstones and had previously undergone ERCP. The median baseline WBC count and CRP values upon presentation to the emergency department were 12,000/mm^3^ (IQR: 9165/mm^3^) and 12.14 mg/dL (IQR: 24.8 mg/dL), respectively. The baseline bilirubin and amylase levels were 2.00 mg/dL and 49.0 UI/L, respectively (Table 1).

Out of the 54 EUS-GBD procedures performed, 24 (44.4%) were cholecysto-gastrostomy (CGS), and 30 (55.4%) were cholecysto-duodenostomy (CDS). EUS-GBD was performed using 20 × 8 mm LAMSs for 10 patients (18.5%), 20 × 10 mm LAMSs for 34 patients (63.0%), and 20 × 16 mm LAMSs for 10 patients (18.5%). A guide wire was used for the placement of the Hot-Spaxus LAMS in 29 out of 54 cases (53.73%), with the intra-channel opening of the proximal flange and release under endoscopic view taking place in 52 out of 54 cases (96.2%). The technical success rate of LAMS placement was 100%. However, due to incomplete biochemical data, specifically data regarding C-reactive protein (CRP) and white blood cell count (WBC) at 5 days after LAMS placement, twenty patients were excluded from the subsequent statistical analysis. Ultimately, clinical success was observed in 23 out of 30 patients, yielding a clinical success rate of 76.67%. We experienced three adverse events, accounting for 5.6% of cases, which were documented in two patients. Specifically, one patient diagnosed with advanced pancreatic cancer experienced a perforation and, regrettably, died due to this complication three days following LAMS placement (Table 2). In another case, stent migration and post-procedural acute pancreatitis were observed. Stent displacement was promptly identified immediately after the procedure and addressed by the placement of a 10 × 10 mm Axios stent. Patients were discharged after a median of five days post-stenting and subsequently monitored via telephone calls for a period of up to 12 months after their discharge from hospital. However, there was a notable dropout rate in the scheduled follow-up telephone calls at the 1, 3, 6, and 12-month intervals following hospital discharge. After one year from stent placement, successful telephone follow-up was achieved for only 10 patients. During the study’s duration, 12 patients passed away. Among these, one patient’s demise was attributed to adverse events following LAMS placement, while only one death was directly linked to unresolved sepsis issues. Data regarding stent removal are available for only 10 patients, with a median removal time of 49.1 days.

## 4. Discussion

In Western countries, gallbladder disease ranks as one of the primary causes of hospitalization, with acute cholecystitis emerging as a predominant complication. According to the Tokyo Guidelines, the conventional management approach for acute cholecystitis involves laparoscopy-assisted cholecystectomy (LAC). However, with the increasing prevalence of elderly populations and the inherent surgical risks associated with this demographic, the exploration of alternative therapeutic avenues has become imperative. Therefore, a valuable therapeutic option has arisen in the form of percutaneous transhepatic gallbladder drainage (PTGBD) or ultrasonography-guided gallbladder drainage (EUS-GBD), both of which aim to alleviate biliary tree pressure through decompression.

A recent meta-analysis conducted by Luk SW et al. demonstrated that EUS-GBD was associated with lower rates of adverse events, shorter hospital stays, and fewer reinterventions and readmissions when compared to PTGBD in patients who were not suitable candidates for surgery [29]. Similarly, Cucchetti et al. conducted a conventional meta-analysis that incorporated a trial sequential analysis of four studies on EUS-GBD and PTGBD in unfit patients with acute cholecystitis. Their analysis revealed statistically higher technical success rates for PTGBD when considering very large sizes, but no statistically significant clinical differences were observed. Similar to the findings of Luk et al., Cucchetti et al. reported that EUS-GBD was linked to lower overall adverse events, reinterventions, and readmissions [30]. Regarding safety concerns, several retrospective studies and a recent network meta-analysis conducted by Podboy and colleagues demonstrated an equivalent rate of adverse events when comparing PTGBD to EUS-GBD [31]. Simultaneously, EUS-GBD exhibited the lowest rates of recurrent cholecystitis when compared to other modalities, such as ETP-GBD and PT-GBD. Presently, European guidelines advocate for the preference of EUS gallbladder drainage over PTGBD in high-surgical-risk patients due to its superior record regarding lower adverse events (AEs) and re-intervention rates [32]. All the aforementioned findings were derived from experiences involving the use of Hot-Axios (Boston Scientific Medical Corporation, Marlborough, MA, USA). Importantly, the present study involved the utilization of a newly dedicated lumen-apposing metal stent, HOT-SPAXUS (Taewoong Medical Co., Gimpo, Republic of Korea), for gallbladder drainage in patients who are unsuitable candidates for surgery due to acute cholecystitis. Successful stent placement was achieved in all fifty-four patients (100%). This achievement appears to be on par with previous experiences involving Hot-Axios [19,29], suggesting that the HOT-SPAXUS device may be both functional and well designed for this specific purpose. In our assessment, we used the EC-LAMS, which is characterized by a length of 2 cm and flexible flanges, two structural elements that significantly contribute to the successful access of the gallbladder. These features prove particularly advantageous when the distance between the gallbladder and the stomach or duodenum is substantial or when these structures are in closer proximity. They enhance the endoscopist’s ability to attain a stable position, thereby enhancing the safety and feasibility of gallbladder drainage [24].

Although the overall results might be promising, three AEs were reported in a small percentage of patients. These adverse events included stent migration, post-procedural acute pancreatitis, and the death of one patient due to a perforation. These rates align with data reported in the literature [19,21,29,33]. It is worth highlighting the absence of stent occlusion and bleeding complications in the study population. It is necessary to carefully monitor and follow-up on patients after EUS-GBD procedures using Hot-Spaxus. In our experience, the clinical success rate was 76.67%. Figure 6 illustrates the rate of decrease in the WBC and PCR analyzed. However, the evaluation of the clinical success of HOT-SPAXUS placement was possible in only 30 out of 54 (55.55%) patients due to a lack of collected data, which certainly influenced the results. Furthermore, it is important to note that the definition of clinical success in our study, which requires the normalization of both clinical and biochemical variables, is more stringent than only the reduction in WBC count as described in some previous studies regarding the use of LAMSs [33]. Finally, the small sample size for the evaluation limited our analysis; on the other hand, the stringent inclusion criteria ensured that our evaluation of the device was not too optimistic. Another important finding is the relatively short length of hospital stay following LAMS placement, which was 5 days, aligning with the recommended duration of antibiotic therapy [14]. This is noteworthy considering the baseline characteristics of the patients, and a limited hospitalization period is a key point worth caring about; the length of stay post-HOT-SPAXUS placement is close to the data reported in the most recent meta-analyses [19,29].

It is crucial to acknowledge the limitations of this study, which include its retrospective nature, the overall sample size, and the limited cases collected from various centers, which may introduce selection bias. Furthermore, the dropout rate in the scheduled follow-up calls and the paucity of data concerning stent removal have constrained our ability to detect instances of cholecystitis recurrence and adverse events associated with stent placement. Moreover, while acute cholecystitis constitutes a significant health concern, particularly among the elderly population, EUS-GBD via the use of Hot-Spaxus LAMS presents a potential alternative treatment option for patients deemed ineligible for surgery. Ongoing and future research endeavors and studies will undoubtedly yield additional insights into the long-term outcomes and efficacy of this approach, thereby contributing to the refinement and optimization of acute cholecystitis management.

## Figures and Tables

**Figure 1 diagnostics-13-03341-f001:**
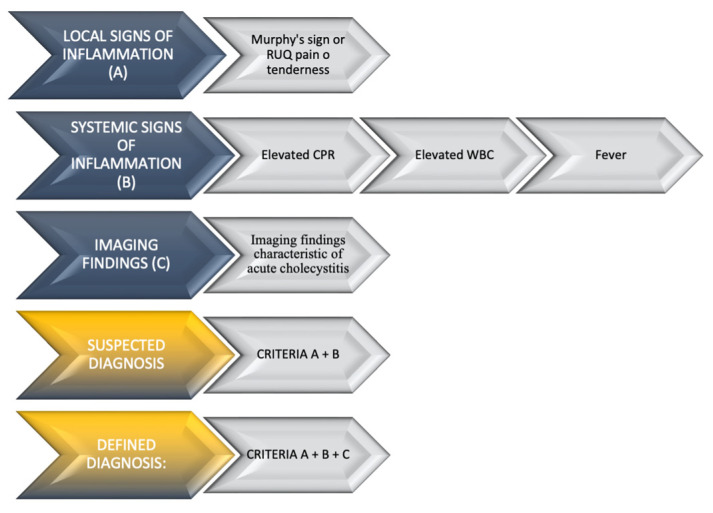
Tokyo Guidelines for acute cholecystitis. Adapted from Yokoe et al. [14].

**Figure 2 diagnostics-13-03341-f002:**
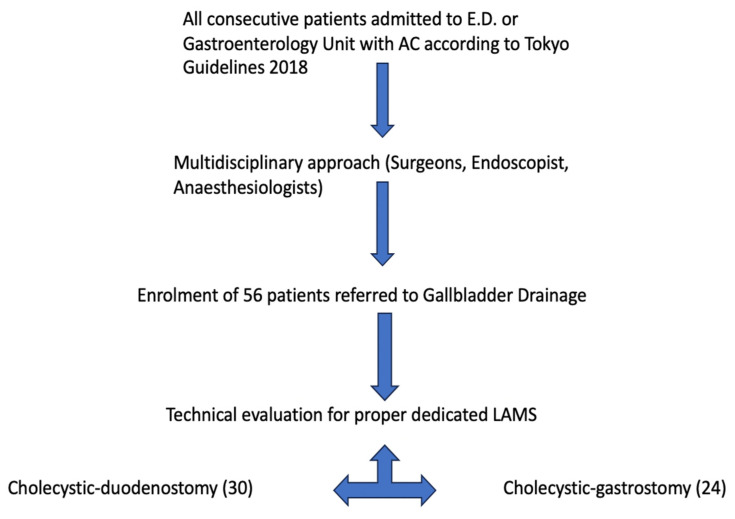
Flow chart detailing the enrollment of the studied patients.

**Figure 3 diagnostics-13-03341-f003:**
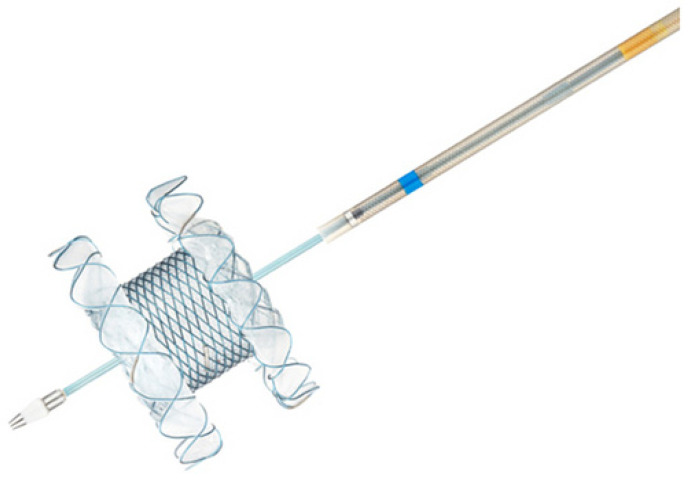
The new electrocautery lumen-apposing metal stent (Hot-Spaxus; Taewoong Medical Co., Ltd., Goyang-si, Republic of Korea).

**Figure 4 diagnostics-13-03341-f004:**
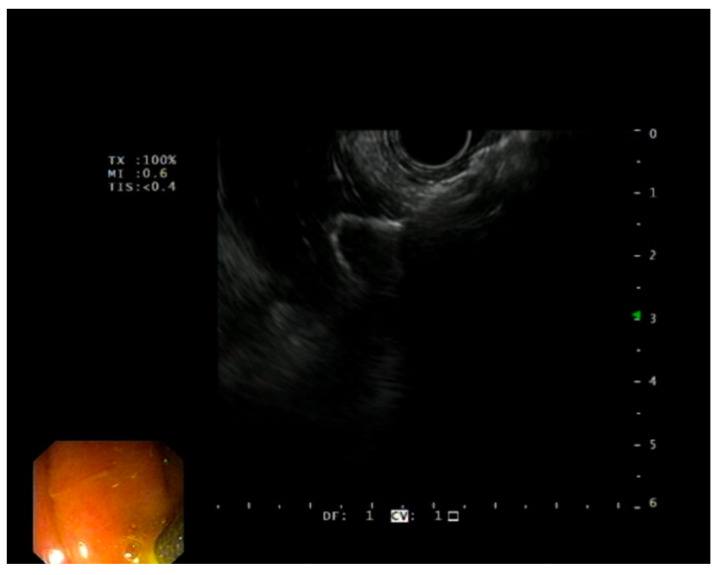
Deployment of distal gallbladder flange under EUS visualization during EUS-GBD.

**Figure 5 diagnostics-13-03341-f005:**
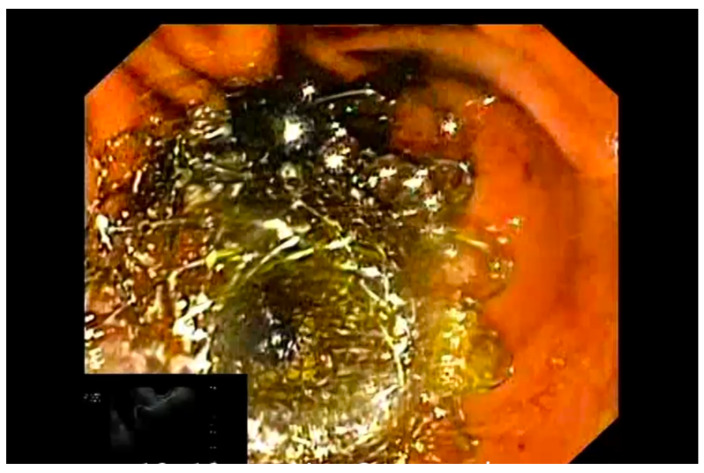
Deployment of proximal gastric flange under endoscopic visualization during EUS-GBD.

**Figure 6 diagnostics-13-03341-f006:**
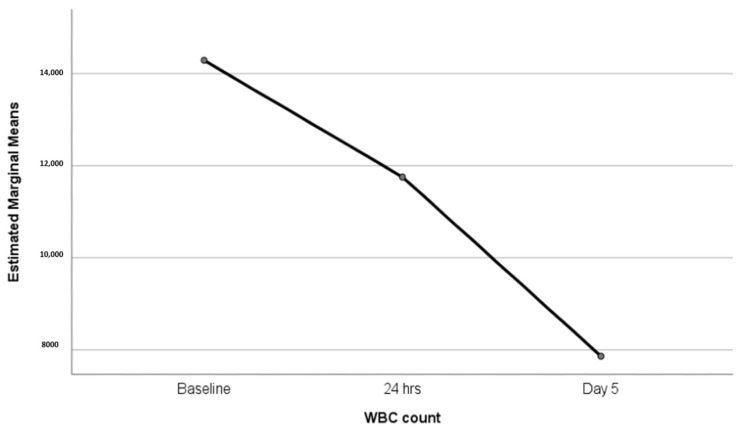
Plot showing trend of WBC count after LAMS placement (*p* < 0.0001).

**Table 1 diagnostics-13-03341-t001:** Outcome parameters.

Parameters	
Technical success	54 (100%)
Clinical success	23/30 (76.67%)
Adverse events (overall)	2 (3.7%)
Perforation	1 (1.9%)
Migration	1 (1.9%)
Acute pancreatitis	1 (1.9%)
Duration of hospital stay after stent placement (days) (median; IQR)	5.0 (5.0)

Numbers are shown in number (%) or median.

**Table 2 diagnostics-13-03341-t002:** Baseline characteristics.

Parameters	
Age (mean ± SD) (years)	76.48 ± 12.6
Male	23 (42.6%)
Previous chemotherapy	16 (29.6%)
CBD stones present	14 (25.9%)
ERCP in same session	5 (9.3%)
Previous ERCP	7 (13.0%)
Baseline liver function tests (median (IQR))	
Total Bilirubin	2.00 (3.91)
Direct Bilirubin	1.63 (2.69)
AST	55.50 (115.25)
ALT	49.0 (74.0)
ALP	223.0 (311.0)
GGT	201.0 (436.50)
WBC count	12,000.0 (9165.0)
CRP	12.14 (24.8)
Amylase	49.0 (57.0)
**Technical details**	
Cholecysto-gastrostomy	24 (44.4%)
Cholecysto-duodenostomy	30 (55.6%)
Stent placed	
20 × 8 mm	10 (18.5%)
20 × 10 mm	34 (63.0%)
20 × 16 mm	10 (18.5%)
Stent placement with guidewire	29 (53.7%)
Intra-channel release	52 (96.3%)

Numbers are shown in number (%) or median.

## Data Availability

Not applicable.

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
