# Peer review of "The Use of a New Dedicated Electrocautery Lumen-Apposing Metal Stent for Gallbladder Drainage in Patients with Acute Cholecystitis"

_diagnostics, 2023, doi:10.3390/diagnostics13213341_

Round 1

Reviewer 1 Report

Dear authors,

We thank you very much for this beautifully designed and written multi-center study. As is known, acute cholecystitis has serious morbidity and mortality in all age groups. Complications, morbidity and mortality are at high rates, especially in elderly patients.

In this study, the high clinical success rate and low side effect rate of ultrasound-guided metal stent placement in a group of patients who are older and unsuitable for surgery provided very beneficial results in patient management in such risky groups.

The publication of this beautiful study in a widely read journal will be very beneficial for the education of both students, assistants and specialists.

We thank you very much for this beautiful work and wish you success in your work.

Author Response

Dear reviewer, 

Thanks for your appreciation. Further investigations will be necessary, LAMS seems to be a valid non invasive therapeutical manovre for unfit or elderly people. Thank you.  

Reviewer 2 Report

Moderate editing requiere

Please correct the following errors

LINE 68      Obstruction by calculi increases intraluminal pressure, inflaming the gallbladder wall and IS FOLLOWED BY bacterial infections, typically enteric organisms like Escherichia coli, Klebsiella spp., and 69 Staphylococcus spp.

LINE 219  Figure 4. Deployment of proximal gastric flange under EUS visualization during EUS-GBD 

 CHANGE FOR  Figure 4. Deployment of proximal gastric flange under ENDOSCOPIC visualization during EUS-GBD  

LINE 223 DELETE he

LINE 234 PLEASE BE SURE THIS SIZES CORRESPOND WITH THE INFORMATION PROVIDED IN LINE 198

LINE 239 EMPLOY THE PAST TENSE Categorical data WAS described using frequency and composition ratios. Continuous data WASdescribed using mean, standard deviation, maximum, minimum, and median

LINE 253 Among the patients who underwent to EUS-GBD, DELETE to

LINE 260 BE SURE THE FOLLOWING SIZES CORRESPOND TO THE INFORMATION PROVIDED PREVIOUSLY IN LINES 198 AND 234.   EUS-GBD was performed using 20 x 8 mm LAMS for 10 patients (18.5%), 20 x 10 mm LAMS for 26 patients (48.1%), 20 x 16 mm LAMS for 10 patients (18.5%) and 10 x 15 mm LAMS for 10 patients 14.8%).

LINE 263 THE FOLLOWING SETENCES ARE COFUSING

A guide wire was used for placement of Hot-Spaxus in 29 out of 54 cases (53.73%) with intrachannel opening of the distal flange and release under endoscopic view in 52 out of 54 cases (96,2%).

WHY USING A GUIDEWIRE IN A FREE-HAND TECHNIQUE? THE INTRACHANNEL OPENING IS OF THE PROXIMAL FLANGE

LINE 265 THE FOLLOWING SENTENCES ARE ALSO CONFUSING

Clinical success was assessed in 30 patients due to not available biochemical data (CRP and WBC, e.g.) atfive days. However, clinical success was achieved in 23 out of 30 patients (76.67%)

Author Response

Dear Pawel, 

As per your suggestion, we have made modifications to our text. Thanks to your input, we carefully reviewed our dataset and identified some errors. I have marked all the corrections for your reference. Regarding the 4000-word text, we believe that adding more words may not be beneficial or necessary. It could risk unnecessary repetition.

Hope this mail find you well. 

Kind regards, 

Luca Brandaleone, MD

Round 2

Reviewer 2 Report

Thank you for the corrections made. It is interesting to have in the endoscopists's armamentarium different kind of devices. In such a way, this lumen apposing metal stent can be another option for gallbladder drainage besides the Axios LAMS.